# Theoretical and Experimental Investigations of a Pseudo-Magnetic Levitation System for Energy Harvesting

**DOI:** 10.3390/s20061623

**Published:** 2020-03-14

**Authors:** Krzysztof Kecik, Andrzej Mitura

**Affiliations:** Department of Applied Mechanics, Lublin University of Technology, 20-618 Lublin, Poland; a.mitura@pollub.pl

**Keywords:** magnetic leviation, energy harvesting, electromagnetic induction, harmonic balance method, experiment

## Abstract

The paper presents an analytical, numerical and experimental analysis of the special designed system for energy harvesting. The harvester system consists of two identical magnets rigidly mounted to the tube’s end. Between them, a third magnet is free to magnetically levitate (pseudo-levitate) due to the proper magnet polarity. The behaviour of the harvester is significantly complicated by a electromechanical coupling. It causes resonance curves to have a distorted shape and a new solution from which the recovered energy is higher is observed. The Harmonic Balance Method (HBM) is used to approximately describe the response and stability of the mechanical and electrical systems. The analytical results are verified by a numerical path following (continuation) method and experiment test with use of a shaker. The influence of harvester parameters on the system response and energy recovery near a main resonance is studied in detail.

## 1. Introduction

### 1.1. Energy Harvesting

Today, more than 40% of all energy consumption is in the form of electrical energy, which is expected to grow to 60% by 2040 [1]. Among the technologies that support the growth of clean energy is the energy harvesting (EH) process technology [2]. The EH is defined as a process where small amounts of energy are recovered from ambient energy and transformed into an electrical energy used immediately (or stored for later usage). An ambient energy is defined as many different forms energy: thermal (e.g., radiation, solar), chemical (e.g., nuclear), electrical, magnetic (e.g., magnetisation), mechanical (e.g., potential, kinetic, elastic, fluid), triboelectric [3,4] and more. The typical energy harvester device comprises one or more transducer mechanisms, power conditioning module, and energy storage (Figure 1).

The transducer mechanism converts energy from one energy type to a another, usually electricity. The power conditioning module is necessary because the output of the transducer can be intermittent or strongly disturbed, and at the wrong frequency, voltage and current to directly drive the device [5]. It rectifies the power and regulates the voltage value. The energy storage unit is needed to the energy accumulate. Usually a rechargeable battery, capacitor, or supercapacitor is used. The power harvested is large enough to power small electronic devices such as sensors [6], structural health monitoring [7,8,9,10] or wireless transmission [11]. The wireless power transfer has been proposed as a new method energy harvesting method. Liu et al. [12] proposed a 5G-based green broadband communication system with simultaneous wireless information and power transfer. A joint optimization unit has been proposed to improve the system performance. In [13], simultaneous cooperative spectrum sensing and energy harvesting model for multi-channel cognitive radio is presented. The obtained results show that this model can achieve higher throughput compared with the traditional. Moreover, the harvested energy can compensate the sensing energy loss and guarantee enough transmission power.

### 1.2. Pseudo-Magnetic Levitation Harvesters

The electromagnetic induction is the most cost effective type of the transducer mechanism [14]. They are based on Faraday’s law of induction—that a change in the magnetic flux of a circuit will result in the induction of an electromotive force. The electromagnetic harvesters are often modelled as a linear devices, then only perform well over a narrow bandwidth close to their natural frequency [15,16,17]. Increasing the damping of the harvester by modifying the electromechanical coupling will extend the bandwidth over which it functions effectively [17,18].

The effective improvement in the operating bandwidth can be achieved by introducing the non-linearity, tuneable resonance, and bi-stability [19,20,21]. The interesting type of the strong nonlinear electromagnetic harvesters use a pseudo-magnetic levitation effect [22,23,24] for energy recovery. Note that magnetic levitation always occurs with a help of a mechanical constraint for stability. The Earnshaw’s theorem proves that it is not possible to achieve magnetic levitation using any combination of the fixed magnets and electric charges. Therefore, the proper term should be the “pseudo-magnetic levitation”. However, in many papers, one can find the term magnetic levitation (maglev) [10,25,26,27] instead of pseudo-magnetic levitation. The pseudo-maglev levitation harvesters are characterized by simplicity of their construction, lack of springs and dampers (not a physical spring that is easily worn out), therefore the time period of usage can be a very long. Moreover, the harvester can recover energy under the external excitation in a low and high frequency ranges. As mentioned earlier, the pseudo-magnetic levitation harvesters are strongly nonlinear systems, therefore are difficult to analyse. Usually, the Finite Element Method has been used to solve the differential equations that govern the dynamics of these systems. Then the magnetic forces, magnetic field distributions are usually easy modelled [25,28]. Mann and Sims [26] presented a design for electromagnetic energy harvesting from the nonlinear oscillations of magnetic levitation. They showed the nonlinear response behaviour of the harvester to be strongly dependent on the damping level. Additionally, they showed interesting aspect that the governing equation for relative displacement reduces to the form of Duffing’s equation. This means that the potential energy function of the maglev oscillator may be expressed as: monostable hardening, monostable softening and bistable. The exemplary potential functions of the Duffing oscillator in Figure 2 are shown. The linear oscillator and nonlinear oscillator which exhibits hardening and softening nonlinearity is monostable. However, for some parameter configurations the oscillator becomes nonlinear bistable, having new stable equilibria.

Their second paper [27] presents analytical solutions for the linear response behaviour of the energy maglev harvester. Their analysis include the inductance coil, but they assumed small value of the electromechanical coupling and the resistance level. They applied the method of undetermined coefficients for analytical analysis. Berdy et al. [29] investigated the power output of a magnetic levitation vibration energy harvester on human participants while they walk and run on a treadmill. The measurement results show that the variation in power generated is relatively significant due to the variation in walking and running gait styles as well as the angle of attachment of the device.

The relevant studies that report the major achievements in the magnetic levitation harvesters in paper [30] are presented. The authors proposed four selection criteria for the harvester comparison: (1) two or more magnets (and one or more coils), (2) including at least one levitating magnet (and one or more fixed magnets), (3) executes axial motions of the levitating magnet(s) and (4) architectures with mono-stable electromagnetic-induction configurations. Moreover, the authors show a comparison of our harvester with other single levitating devices.

This paper presents complex analysis of the single pseudo-levitating harvester which is solved analytically by the HBM, numerically by the continuation method (CM) and verified by experimental shaker tests. In the literature, the numerical and experimental studies of the maglev harvesters were used. We propose the approximate analytical method to find an analytical solutions, and show analytical influence of the system parameters on energy harvesting. Additionally, the stability analysis of obtained analytical solutions is performed. All analytical results show a good compliance with the numerical research and an experiment.

## 2. Energy Harvesting System Design

### 2.1. Magnetic Levitation Architectures

The analyzed single pseudo-levitating magnet harvester plays role of the pendulum tuned mass damper. It is mounted on a special laboratory harvester–absorber system, which allows for simultaneous vibration reduction and energy recovery. More information about this can be found in [31,32].

The single levitating harvester in Figure 3a is presented. It consists of the cylindrical non-magnetic tube with the two fixed ring permanent magnets (Figure 3b).

The ring magnets have 20 mm in diameter and 10 mm in height. The levitating (moving) cylindrical permanent neodymium magnet of 20 mm in diameter and 30 mm in height is placed inside the tube between the fixed ring magnets. It motion is limited by the repulsive force exerted by the opposing magnetic field of magnets placed. The teflon surface was applied around the moving magnet to reduce friction between the moving magnet and the inner tube’s surface. Additionally, special air holes and gaps to air compression reduction are made. To prevent the magnet’s impact special rubber bumpers are installed. The induction coil is wrapped around outside of the tube, and is wound from 140 μm diameter Copper wire. The coil has 12,740 wires, the inductance of 1.46 H and the own resistance of 1.15 kΩ. The separation distance between the moving and the bottom fixed magnet was 0.041 m. It can be changed by the screw system. This allows a modification in the magnetic suspension parameters. The levitating magnet moving axially through the center of a coil will induce a voltage across the coil terminals. The practical application of the magnet coil system can be shaker flashlights. If the flashlights vigorously shaken back and forth, then a magnet to move through a coil causing charge to the battery [33].

### 2.2. Mathematical Model

The dynamics of the pseudo-maglev harvester can be modeled as the nonlinear spring-mass-damper mechanical system (Figure 4a) with an external base excitation y(t), connected to the electrical circuit (Figure 4b). The circuit consists of the coil with the inductance (*L*) and the own ($R_{C}$) and load ($R_{L}$) resistances. The sum of $R_{L}$ and $R_{C}$ the total resistance (*R*) is called. When the magnet moves inside the coil, then electromagnetic induction of voltage U(t) and current i(t) flow in the electrical circuit occurs. The electrical circuit dissipates the produced energy across a load resistor (the recovered current is converted into heat across the load resistor).

Defining a new coordinate z(t)=x(t)−y(t) which means the relative displacement between the vibrating structure and the moving magnet, the equations of motion can be written:(1)$$m\ddot{z}\left(t\right)+c\dot{z}\left(t\right)+kz\left(t\right)+k_{1}z^{3}\left(t\right)+\alpha i\left(t\right)+mg=mQ\omega^{2}cos\left(\omega t\right),$$
(2)$$L\dot{i}\left(t\right)+\left(R_{L}+R_{C}\right)i\left(t\right)=\alpha \dot{z}\left(t\right).$$

The pseudo-maglev suspension is modelled as a magnetic spring with the stiffness components *k* and $k_{1}$ (Duffing type) and the linear dashpot described by the coefficient *c*. A cubic nonlinearity comes from the magnetic restoring forces between the magnets [26,27,32]. The parameter *m* is the magnet mass, while *g* is the gravitational acceleration. The excitation is assumed to be harmonic y(t)=Qcos(ωt). The electrical and mechanical systems are coupled by the electromechanical coupling coefficient α. It characterizes how the induced voltage is related to the velocity of the magnet. The terms αi(t) and $\alpha \dot{z}\left(t\right)$ are called electrodynamic Lorentz force (edf) and the well known electromotive force (emf), respectively. The fundamental relations for electromechanical systems is that emf = edf [26].

In the literature, different models of α can be found [25,34,35]. Generally, this parameter depends on the coil design and is a function of the magnet’s position versus the coil [18,34]. However in paper [32] authors show that fixed value of the coupling coefficient can be accepted, provided that it is properly chosen. They showed comparison between classical (fixed value) and novel (nonlinear polynomial) electromechanical coupling modelling. Moreover, in the literature the coil inductance is assumed to be very small and finally is neglected. This leads that the electrical current flow equation is reduced [36] and the electromechanical coupling coefficient is part of damping called electromagnetic.

### 2.3. Shaker Test

The laboratory experiments have been performed using the electromagnetic shaker system TIRAvib50101 which in Figure 5 is shown. The shaker was controlled by LMS.TestLab software which provided the sinusoidal input excitation signal during vibration tests and reproduced environmental conditions over a required frequency band. The two acceleration sensors for measuring the magnet’s response and to control of the excitation have been applied. Measurements of the shaker acceleration were obtained by mounting a sensor to the shaker surface. Measurements of the centre moving magnet acceleration were obtained by a second accelerometer mounted on the special element glued to the magnet.

The vibration signals are recorded in LMS SCADAS and the data is processed with LMS.Test.Lab software to obtain the acceleration and displacement spectres. The special designed harvester module consists of the DSP MicroDAQ module with multi-core application processor OMAP L137, the amplifiers, and the load resistance module which allowed to modify the resistance. The experimental results have been formed by an upward and downward frequency sweep.

## 3. Harmonic Balance Method (HBM)

### 3.1. Stability Analysis

Because the equations of motion (Equation 1) and (Equation 2) include nonlinear terms it is difficult to find their strictly correct solutions. Therefore, in the neighbourhood of main resonance the approximate solutions by the harmonic balance method (HBM) are sought. Generally, the HBM is an approximate analytical method for the study of non-linear systems described by ODE equations. The main essence is to replace the nonlinear parts the by specially-constructed functions to find approximate solutions of the nonlinear systems. To verify the accuracy of the HBM, numerical simulations have been used. The numerical simulation based on the numerical continuation technique in Auto07p software [37]. This method gives a deeper understanding of the solution behaviour: stability, multiplicity and bifurcations.

The pseudo-maglev harvester response has a predominant basic frequency ω under the external periodic excitation. Therefore, the first approximate solutions are assumed as follows:(3)$$z\left(t\right)=z_{0}\left(t\right)+A\left(t\right)sin\left(\omega t\right)+B\left(t\right)cos\left(\omega t\right),$$
(4)i(t)=C(t)sin(ωt)+D(t)cos(ωt),
where A(t), B(t), C(t), D(t) are unknown amplitudes of the harvester and amplitudes of the induced current. The amplitude $z_{0}$ is responsible for the vibration centre shift of the levitating magnet.

After substituting Equations (Equation 3) and (Equation 4) into Equations (Equation 1) and (Equation 2), and then balancing the coefficients of the corresponding sine and cosine terms we get a set of first-order approximate differential modulation equations. For better clarity, assumed notations A(t)≡A, B(t)≡B, C(t)≡C, D(t)≡D and $z_{0}\left(t\right)\equiv z_{0}$.
(5)$$\begin{array}{c} {\dot{z}}_{0}c+mg+z_{0}\left(k+\frac{3}{2}A^{2}k_{1}+\frac{3}{2}B^{2}k_{1}+k_{1}z_{0}^{2}\right)=0, \end{array}$$
(6)$$\begin{array}{c} c\left(\dot{B}+A\omega \right)+m\left(2\dot{A}\omega -B\omega^{2}\right)+Bk+\frac{3}{4}A^{2}Bk_{1}+\frac{3}{4}B^{3}k_{1}-mQ\omega^{2}+3Bk_{1}z_{0}^{2}+D\alpha =0, \end{array}$$
(7)$$\begin{array}{c} c\left(\dot{A}-B\omega \right)-m\left(2\dot{B}\omega +A\omega^{2}\right)+Ak+\frac{3}{4}A^{3}k_{1}+\frac{3}{4}AB^{2}k_{1}+3Ak_{1}z_{0}^{2}+C\alpha =0, \end{array}$$
(8)$$\begin{array}{c} L\left(\dot{D}+C\omega \right)-\alpha \left(\dot{B}+A\omega \right)+DR=0, \end{array}$$
(9)$$\begin{array}{c} L\left(\dot{C}-D\omega \right)-\alpha \left(\dot{A}-B\omega \right)+CR=0. \end{array}$$

Stability analysis of the harmonic solutions is carried out by using the approximate Equations (Equation 5)–(Equation 9). Determining derivatives $\dot{A},\dot{B},\dot{C},\dot{D}$ and ${\dot{z}}_{0}$, we get the so-called amplitude modulation equations which can be written:(10)$$\begin{array}{c} {\dot{z}}_{0}=-\frac{3k_{1}A^{2}z_{0}+3k_{1}B^{2}z_{0}+2k_{1}z_{0}^{3}+2kz_{0}+2gm}{2c}, \end{array}$$
(11)$$\begin{array}{c} \dot{A}=\frac{-6k_{1}A^{2}Bm\omega -3k_{1}AB^{2}c-24k_{1}Bz_{0}^{2}m\omega +4Bc^{2}\omega +8Bm^{2}\omega^{3}-8kBm\omega -6k_{1}B^{3}m\omega }{4(c^{2}+4m^{2}\omega^{2})}\\ \frac{-3k_{1}A^{3}c-12k_{1}Az_{0}^{2}c-4Acm\omega^{2}-4kAc-4C\alpha c+8Qm^{2}\omega^{3}-8D\alpha m\omega }{4(c^{2}+4m^{2}\omega^{2})}, \end{array}$$
(12)$$\begin{array}{c} \dot{B}=\frac{6k_{1}A^{3}m\omega -3k_{1}A^{2}Bc+6k_{1}AB^{2}m\omega +24k_{1}Az_{0}^{2}m\omega -4Ac^{2}\omega -8Am^{2}\omega^{3}+8kAm\omega }{4(c^{2}+4m^{2}\omega^{2})}\\ \frac{-3k_{1}B^{3}c-12k_{1}Bz_{0}^{2}c-4Bcm\omega^{2}-4kBc+4Qcm\omega^{2}-4D\alpha c+8C\alpha m\omega }{4(c^{2}+4m^{2}\omega^{2})}, \end{array}$$
(13)$$\begin{array}{c} \dot{C}=\frac{-3k_{1}A^{3}\alpha c-6k_{1}A^{2}B\alpha m\omega -3k_{1}AB^{2}\alpha c-12k_{1}Az_{0}^{2}\alpha c-4A\alpha cm\omega^{2}-4kA\alpha c-6k_{1}B^{3}\alpha m\omega }{4L(c^{2}+4m^{2}\omega^{2})}\\ \frac{-24k_{1}Bz_{0}^{2}\alpha m\omega -8B\alpha m^{2}\omega^{3}-8kB\alpha m\omega -4C\alpha^{2}c-8D\alpha^{2}m\omega +8Q\alpha m^{2}\omega^{3}+4DLc^{2}\omega }{4L(c^{2}+4m^{2}\omega^{2})}\\ \frac{-4CRc^{2}+16DLm^{2}\omega^{3}-16CRm^{2}\omega^{2}}{4L(c^{2}+4m^{2}\omega^{2})}, \end{array}$$
(14)$$\begin{array}{c} \dot{D}=\frac{6k_{1}A^{3}\alpha m\omega -3k_{1}A^{2}B\alpha c+6k_{1}AB^{2}\alpha m\omega -4kB\alpha c+8A\alpha m^{2}\omega^{3}+8kA\alpha m\omega -3k_{1}B^{3}\alpha c}{4L(c^{2}+4m^{2}\omega^{2})}\\ \frac{24k_{1}Az_{0}^{2}\alpha m\omega -12k_{1}Bz_{0}^{2}\alpha c-4B\alpha cm\omega^{2}-4D\alpha^{2}c+8C\alpha^{2}m\omega +4Q\alpha cm\omega^{2}-4CLc^{2}\omega }{4L(c^{2}+4m^{2}\omega^{2})}\\ \frac{-4DRc^{2}-16CLm^{2}\omega^{3}-16DRm^{2}\omega^{2}}{4L(c^{2}+4m^{2}\omega^{2})}. \end{array}$$

To determined stability of the obtained solutions is based on the Jacobian eigenvalues analysis, where the Jacobian takes the form
(15)$$\begin{array}{c} \left|\begin{array}{ccccc} \frac{\partial {\dot{z}}_{0}}{\partial z_{0}}-\lambda & \frac{\partial {\dot{z}}_{0}}{\partial A} & \frac{\partial {\dot{z}}_{0}}{\partial B} & \frac{\partial {\dot{z}}_{0}}{\partial C} & \frac{\partial {\dot{z}}_{0}}{\partial D}\\ \frac{\partial \dot{A}}{\partial z_{0}} & \frac{\partial \dot{A}}{\partial A}-\lambda & \frac{\partial \dot{A}}{\partial B} & \frac{\partial \dot{A}}{\partial C} & \frac{\partial \dot{A}}{\partial D}\\ \frac{\partial \dot{B}}{\partial z_{0}} & \frac{\partial \dot{B}}{\partial A} & \frac{\partial \dot{B}}{\partial B}-\lambda & \frac{\partial \dot{B}}{\partial C} & \frac{\partial \dot{B}}{\partial D}\\ \frac{\partial \dot{C}}{\partial z_{0}} & \frac{\partial \dot{C}}{\partial A} & \frac{\partial \dot{C}}{\partial B} & \frac{\partial \dot{C}}{\partial C}-\lambda & \frac{\partial \dot{C}}{\partial D}\\ \frac{\partial \dot{D}}{\partial z_{0}} & \frac{\partial \dot{D}}{\partial A} & \frac{\partial \dot{D}}{\partial B} & \frac{\partial \dot{D}}{\partial C} & \frac{\partial \dot{D}}{\partial D}-\lambda \end{array}\right|=0. \end{array}$$

The stability of the obtained solutions depends on the roots of λ. If all roots have the negative real part then the solution is stable.

### 3.2. Approximate Analytical Solutions

For a steady state, amplitudes are constant thus the first-order derivatives are equal to zero
(16)$$\begin{array}{c} {\dot{z}}_{0}=0,\dot{A}=0,\dot{B}=0,\dot{C}=0,\dot{D}=0. \end{array}$$

Introducing Equation (Equation 16) into Equations (Equation 5)–(Equation 9) and after few mathematical manipulations we obtain the following algebraic equations:(17)$$\begin{array}{c} mg+z_{0}\left(k+\frac{3}{2}A^{2}k_{1}+\frac{3}{2}B^{2}k_{1}+k_{1}z_{0}^{2}\right)=0, \end{array}$$
(18)$$\begin{array}{c} Bk+\frac{3}{4}A^{2}Bk_{1}+\frac{3}{4}B^{3}k_{1}-mQ\omega^{2}+Ac\omega -Bm\omega^{2}+3Bk_{1}z_{0}^{2}+D\alpha =0, \end{array}$$
(19)$$\begin{array}{c} Ak+\frac{3}{4}A^{3}k_{1}+\frac{3}{4}AB^{2}k_{1}-Bc\omega -Am\omega^{2}+3Ak_{1}z_{0}^{2}+C\alpha =0, \end{array}$$
(20)$$\begin{array}{c} DR+CL\omega -A\omega \alpha =0, \end{array}$$
(21)$$\begin{array}{c} CR-DL\omega +B\omega \alpha =0. \end{array}$$

Note that Equations (Equation 20)–(Equation 21) are linear. Equations (Equation 17)–(Equation 21) have been solved with help of the Mathematica software. After a few mathematical manipulations, the amplitudes *C* and *D* are:(22)$$\begin{array}{c} C=\frac{\omega (-BR+AL\omega )\alpha }{R^{2}+L^{2}\omega^{2}}, \end{array}$$
(23)$$\begin{array}{c} D=\frac{\omega (AR+BL\omega )\alpha }{R^{2}+L^{2}\omega^{2}}. \end{array}$$

Then Equations (Equation 22) and (Equation 23) are squared and added, where $z_{1}^{2}$ = $A^{2}$+$B^{2}$. This leads to the fact that current *i* ($i^{2}$ = $C^{2}$+$D^{2}$) and the recovered power $P=Ri^{2}$ responses are written in the simple form:(24)$$\begin{array}{c} i=\frac{\omega \alpha z_{1}}{\sqrt{R^{2}+L^{2}\omega^{2}}},P=\frac{\omega^{2}\alpha^{2}z_{1}^{2}R}{R^{2}+L^{2}\omega^{2}}. \end{array}$$

Introducing amplitudes *C* and *D* (Equations (Equation 22) and (Equation 23)) into Equations (Equation 18) and (Equation 19), we get the amplitudes *A* and *B*:(25)$$\begin{array}{c} A=-\frac{mQ\omega^{3}\left(-c-\frac{R\alpha^{2}}{R^{2}+L^{2}\omega^{2}}\right)}{\frac{{\left(cR^{2}\omega +cL^{2}\omega^{3}+R\omega \alpha^{2}\right)}^{2}}{{\left(R^{2}+L^{2}\omega^{2}\right)}^{2}}+{\left(k+\frac{3}{4}k_{1}\left(z_{1}^{2}+4z_{0}^{2}\right)+\omega^{2}\left(-m+\frac{L\alpha^{2}}{R^{2}+L^{2}\omega^{2}}\right)\right)}^{2}}, \end{array}$$
(26)$$\begin{array}{c} B=\frac{mQ\omega^{2}\left(k+\frac{3}{4}k_{1}\left(z_{1}^{2}+4z_{0}^{2}\right)+\omega^{2}\left(-m+\frac{L\alpha^{2}}{R^{2}+L^{2}\omega^{2}}\right)\right)}{\frac{{\left(cR^{2}\omega +cL^{2}\omega^{3}+R\omega \alpha^{2}\right)}^{2}}{{\left(R^{2}+L^{2}\omega^{2}\right)}^{2}}+{\left(k+\frac{3}{4}k_{1}\left(z_{1}^{2}+4z_{0}^{2}\right)+\omega^{2}\left(-m+\frac{L\alpha^{2}}{R^{2}+L^{2}\omega^{2}}\right)\right)}^{2}}. \end{array}$$

Finally, the sixth-order non-linear polynomial equation describing the magnet’s amplitude and the magnet vibration centre shift are obtained:(27)$$\begin{array}{c} \left[9k_{1}^{2}\left(R^{2}+L^{2}\omega^{2}\right)\right]z_{1}^{6}+24k_{1}\left(R^{2}+L^{2}\omega^{2}\right)\left[k-m\omega^{2}+3k_{1}z_{0}^{2}+\frac{L\omega^{2}\alpha^{2}}{R^{2}+L^{2}\omega^{2}}\right]z_{1}^{4}+\\ +16\left(R^{2}+L^{2}\omega^{2}\right)\left[k^{2}+c^{2}\omega^{2}-2km\omega^{2}+m^{2}\omega^{4}+6kk_{1}z_{0}^{2}-6k_{1}m\omega^{2}z_{0}^{2}+9k_{1}^{2}z_{0}^{4}\right]z_{1}^{2}+\\ +16\omega^{2}\alpha^{2}\left[2kL+2cR-2Lm\omega^{2}+6k_{1}Lz_{0}^{2}+\alpha^{2}\right]z_{1}^{2}-\\ -16m^{2}Q^{2}\omega^{6}\left(R^{2}+L^{2}\omega^{2}\right)=0, \end{array}$$
(28)$$\begin{array}{c} mg+z_{0}\left(k+\frac{3}{2}k_{1}z_{1}^{2}+k_{1}z_{0}^{2}\right)=0. \end{array}$$

The frequency–amplitude harvester response is given by solving of Equations (Equation 27) and (Equation 28).

## 4. Methods

In order to check the correctness of the HBM method, the obtained results were compared with the numerical continuation method (CM) and the experiment. The CM is a very good numerical method suited for tracing one-dimensional manifolds, curves (called branches) of solutions. It allows to detect the bifurcation points, new solution branches and calculate the stable and unstable solutions. The theoretical analysis is based on the parameters from the experimental laboratory rig (Figure 5). Some of these are readily measurable: *m* = 0.09 kg, *L* = 1.46 H, $R_{L}$ = 0 ÷ 10 kΩ, $R_{C}$ = 1.15 kΩ and *Q* = 0.6 mm. The damping coefficient is estimated from the logarithmic decrement method equals *c* = 0.07 Ns/m. In order to determine the magnetic suspension parameters, the force–displacement relationship was conducted with use of Shimadzu tensile tests. Of course, the linear stiffness depends on the distance between the fixed and moving magnets [34]. In our case, these parameters were *k* = 640 N/m and $k_{1}$ = 460 kN/m${}^{3}$. The value of $k_{1}$ depends on geometry and size of the all magnets. The distance between the moving magnet (with sensor) and the shaker was 0.15 m. It has been estimated based on our own tests and suggestions from [38].

## 5. Results and Discussion

### 5.1. Single Pseudo-Levitating Magnet Vibration Centre Shift

The magnet-coil configuration influences the electromechanical coupling [32]. Therefore, the magnet vibration centre must be taken into consideration. The detail analysis of the Equation (Equation 28) shows that the magnet vibration centre $z_{0}$ depends on the magnet oscillation amplitude $z_{1}$.

In Figure Equation 6a, we can see that the magnet amplitude $z_{1}$ influences the vibration center shift. At high enough oscillation amplitudes, the moving magnet vibration centre shift is practically independent from the linear stiffness (Figure 6a).

Moreover, it can be seen that for $z_{1}$ = 0 we observe the static displacement of the moving magnet. Of course, an increase of the linear stiffness component *k* reduces the static displacement of magnet.

Figure 6b depicts the magnet amplitude influencing the vibration centre shift. The analysis was done for three different nonlinear stiffness: $k_{1}$ = 100 kN/m${}^{3}$ (red line), $k_{1}$ = 460 kN/m${}^{3}$ (black line) and $k_{1}$ = 1000 kN/m${}^{3}$ (blue line). For the low magnet oscillation, the influence of the nonlinearity on the vibration centre is weak. An increase of the oscillation amplitude causes that the vibration centre is reduced.

### 5.2. Analytical Results

In order to show the influence of the amplitude excitation (*Q*), a series of analyses have been performed. The exemplary resonance curves of the moving magnet and the recovered power versus the frequency ω in Figure 7 are shown. The blue line corresponds with the case where *Q* = 0.1 mm, the red line to *Q* = 0.5 mm, the green line to *Q* = 0.8 mm and the black line to *Q* = 1.1 mm.

The most effective energy harvesting occurs in the resonance conditions. The resonance peak is located close to the frequency ω≈ 88 rad/s. Of course, a greater magnet oscillation amplitude *Q* causes an increase in the magnet response and leads to higher level of the recovered power. For example, an increase of the excitation amplitude about two times (from 0.6 mm to 1.1 mm) causes a two times increase in the magnet vibration and a four times increase in the recovered power. Note that all resonance curves are stable and have the linear behaviour.

The frequency response curves of the pseudo-maglev harvester under the total resistance *R* in Figure 8 are shown. The analytical study has been done for the resistances: 0.5 kΩ (blue line), 1 kΩ (red line), 2 kΩ (green line) and 5 kΩ (black line). Upon analysing and comparing the obtained results from both diagrams, we can conclude that the resistance strongly influences the magnet dynamics and energy harvesting.

An increase of *R* causes the magnet oscillation to become higher. This effect can be explained by analysis of Equations (Equation 1) and (Equation 2), and neglecting the inductance (*L* = 0 H). Then, edf force (Equation (Equation 1)) can be written as edf = $\frac{\alpha^{2}\dot{z}}{R}$. This means that the higher resistance decreases the electrical damping level (edf force). However, the EH under *R* looks quite different. A low value of *R* causes the resonance peak not to be observed. An increase of *R* causes higher magnet oscillation but reduces the effective bandwidth of the power (Figure 8b).

The parameter *L* (inductance) characterizes behaviour of the coil. It is defined in terms of that opposing emf force or its generated magnetic flux and the corresponding electric current. The coil inductance depends on the geometry of the current path as well as the magnetic permeability of nearby materials. As mentioned earlier, much of the literature suggests that the coil inductance does not influence the frequency response as well as energy harvesting. This is true only for the low inductance (usually typical in real practice). Our analytical results (Figure 9a,b) show that the inductance lower than 1H can be neglected.

A main question seems to be when the *L* can be ignored. In this aim, the relationship between the moving magnet response (Figure 10a) and the recovered current (Figure 10b) versus the coil inductance has been plotted. For the analysis, three different resistance levels: *R* = 1 kΩ (red line), *R* = 2 kΩ (black line) and *R* = 5 kΩ (blue line) have been selected. Assuming, that the neglect of *L* causes a 2% error in the magnet’s oscillation, then the maximal coil inductance can be: 1.28 H for *R* = 1 kΩ, 1.69 H for *R* = 2 kΩ and 2.57 H for *R* = 5 kΩ (Figure 10a). Whereas, for a 2% error in the recovered current, the inductance can range: 1.84 H for *R* = 1 kΩ, 2.07 H for *R* = 2 kΩ and 2.25 H for *R* = 5 kΩ (Figure 10b). The analysis clearly shows that the coil inductance can be ignored, but this depends on the total resistance level.

The last analysed parameter is the electromechanical coupling coefficient (α). This parameter characterizes how the induced current in the coil is related to the velocity of the moving magnet [32]. As shown in the amplitude-frequency, plot this parameter is a crucial from the dynamics (Figure 11a) and energy harvesting (Figure 11b) point of view. We can see that the electromechanical coupling coefficient strongly affects the magnet amplitude responses. If the electromechanical coefficient increases, then the magnet’s amplitude and the recovered energy are significantly reduced. This means that α can be treated as the electrical damping coefficient (increases edf force in Equation (Equation 1)).

Additionally, the jump amplitude phenomena (well-known as the foldover effect) close to the main resonance is observed. For higher values of α the jump phenomena gradually disappears. The jump phenomena causes coexistence of two stable and one unstable solutions and is positive for EH. One of the responses is characterized by the higher energy input (from top branch). For example, for α = 12 V/A the maximal recovered power from top branch equals about *P* = 2.7 W while from the bottom branch is *P* = 0.03 W (Figure 11b). Moreover, the foldover effect causes a broadening effective frequency bandwidth. For the coupling value of α = 20 V/A the effective frequency bandwidth is located ω≈ 82 ÷ 92 rad/s, but for α = 12 V/A the region became wider to ω≈ 85 ÷ 125 rad/s. Of course, the higher EH is strongly related with increasing of the magnet oscillation (Figure 11a).

### 5.3. Experimental Verification

A series of experimental tests for the two different resistances have been performed: *R* = 2 kΩ and *R* = 10 kΩ. The exemplary resonance curves of the moving magnet response ($z_{1}$) and the recovered current (*i*) in Figure 12 and Figure 13 are shown respectively. The experiment tests are marked by the red plus signs. The blue line means the numerical stable solution, while the blue circle points denote the stable solution obtained by the HBM.

The HBM and CM results are compliance in the total frequency range. This means that the first approximation of the assumed solutions is sufficient. The experimental results are similar to the analytical and numerical. A few small differences probably come from the magnet-tube friction. The maximal recovered current has been obtained close to the resonance peak ω = 88 rad/s equals *i* = 5.8 mA (what correspond power *P* = 0.067 W).

Modification in the coupling coefficient and the resistance values cause the resonance curves to have a distorted shape and the foldover effect to appear (Figure 13a,b).

The blue lines mean the stable numerical solution, the blue circle points denote the stable solution obtained by the HBM, the red lines mean the unstable numerical solution, and the red point show the unstable analytical solution. The experimental verification marked again by the red plus signs. The analytical and numerical results are in compliance. The foldover effect exhibits two stable and one unstable solutions has been confirmed experimentally. The numerical recovered power from the top branch equals *P* = 3.6 W, while about *P* = 0.036 W from the bottom branch.

The exemplary experimental and numerical recovered time histories in Figure 14 are shown. We can see a current flow in the electrical circuit for two different electrical parameters configurations.

The red color means the recovered current from the pseudo-maglev harvester, while the black line means numerical result. The numerical time histories are clear periodic signals, while the experimental signals are slightly disturbed probably by the nonlinear friction between the levitating magnet and tube, the model simplification and parameters identification. However, both signals have similar periods and amplitudes.

## 6. Conclusions

The paper presents theoretical and experimental analysis of the single pseudo-levitating harvester. The HBM method is successfully applied to obtain the analytical responses and calculation of the stability solutions. The analytical, numerical and experimental results are in compliance already for the first assumed approximation solutions. Moreover, the influence of the electrical parameters on energy harvesting is investigated in detail.

The obtained results show that the magnet vibration centre shift depends on the magnet’s oscillation amplitude. This is very a important conclusion because the magnet shift relative to the coil changes the electromechanical coupling value.

The frequency response strongly depends on the resistance and electromechanical coupling coefficient. An increase in the resistance causes the higher oscillation of the levitating magnet. Moreover, the coupling coefficient strongly influences the resonance curve shape. In low values of α, the strong nonlinear behaviour is observed with two stable periodic solutions. The obtained results confirm that the coil inductance can be neglected; however, this depends on the resistance level. For the analyzed parameters, the coil inductance can be neglected if is smaller then 1.28 H. The maximum recovered current from numerical tests was 3.6 W, while from the experiment it was about 1 W.

The obtained results show that for the tested device the recovered power reaches few watts. It is expected that in larger structures, the harvested energy will get higher amounts. The effectiveness of the systems is relatively small—about 2% of input. However, the proposed device is a part of the pendulum absorber/harvester system. Therefore, the recovered current mainly will come from the vibration mitigation.

A more detailed experimental investigation and the shaping of the electromechanical coupling by the special designed magnet-separators stacks and the energy leakage problem will be carried out in a future work.

## Figures and Tables

**Figure 1 sensors-20-01623-f001:**
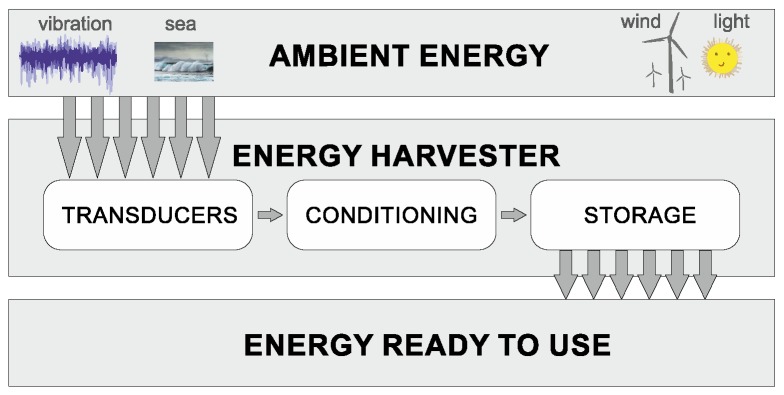
Typical energy harvesting process.

**Figure 2 sensors-20-01623-f002:**
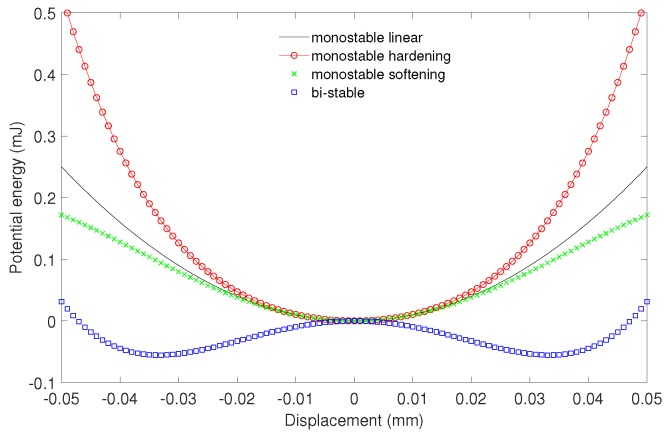
The potential energy function of the Duffing oscillator.

**Figure 3 sensors-20-01623-f003:**
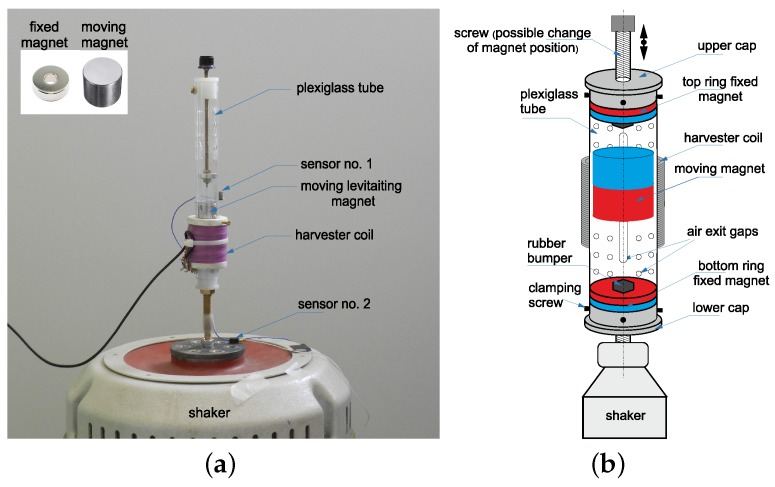
Photo (**a**) and scheme of the single pseudo-levitating harvester (**b**). This system was originally designed and applied as the pendulum tuned mass damper (see paper [31]).

**Figure 4 sensors-20-01623-f004:**
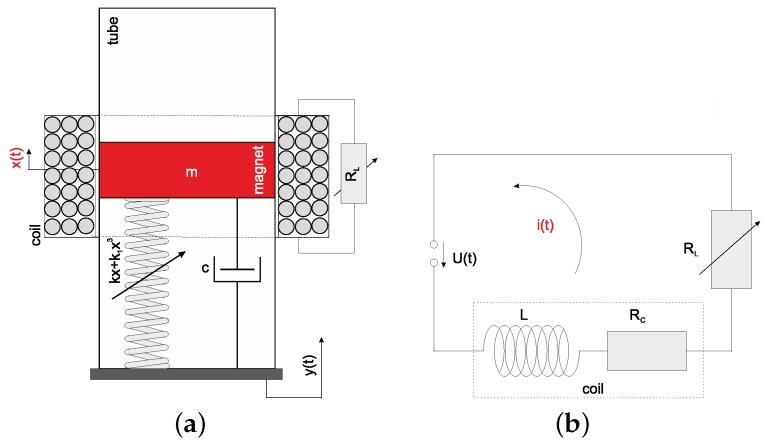
Physical model of pseudo-maglev harvester (**a**) and the electrical circuit comprising of the coil and the external load resistor (**b**). The magnet oscillation causes the current induction in the electrical circuit.

**Figure 5 sensors-20-01623-f005:**
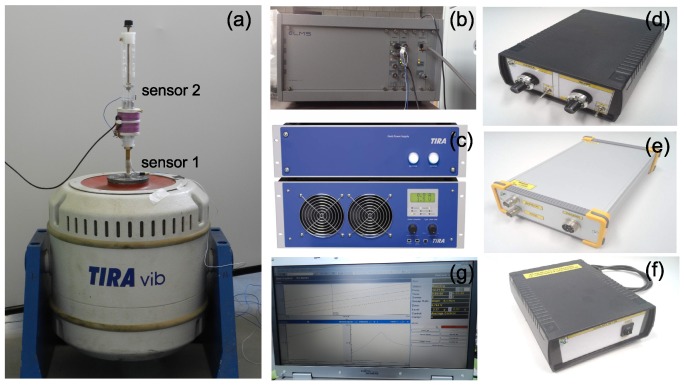
Measuring stand: shaker TIRA vib 50101 with the harvester (**a**), the LMS SCADAS module (**b**) with the analogue TIRA amplifiers (**c**), the load resistance module (**d**), the harvester DSP module (**e**), the harvester amplifiers (**f**) and the PC with the TestLab software (**g**).

**Figure 6 sensors-20-01623-f006:**
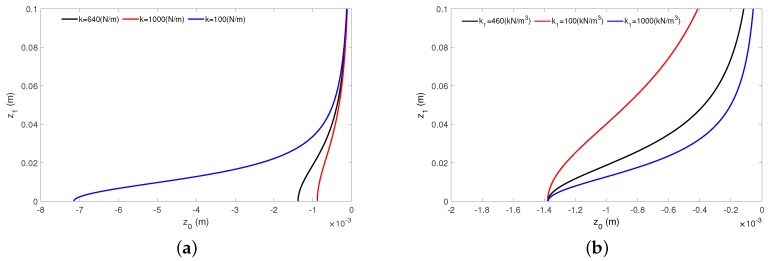
An influence of the linear (**a**) and nonlinear (**b**) stiffness components on the magnet vibration centre. The diagrams have been obtained for parameters $k_{1}$ = 460 kN/m${}^{3}$ and $k_{1}$ for *k* = 640 N/m from the Equation (Equation 28).

**Figure 7 sensors-20-01623-f007:**
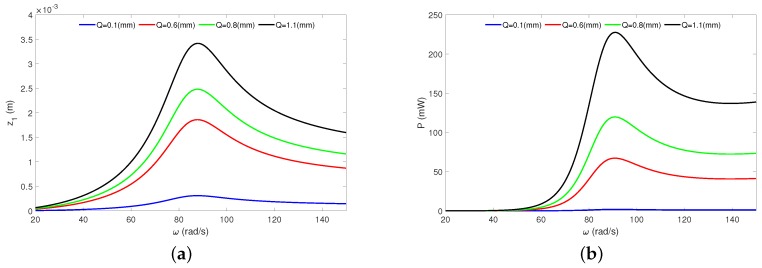
Analytical HBM results. The moving magnet’s resonance curves (**a**) and the recovered power (**b**) under the external periodic excitation. The results have been calculated for the resistance R = 2.0 kΩ. The results have obtained from Equations (Equation 27) and (Equation 28). All resonance responses are stable.

**Figure 8 sensors-20-01623-f008:**
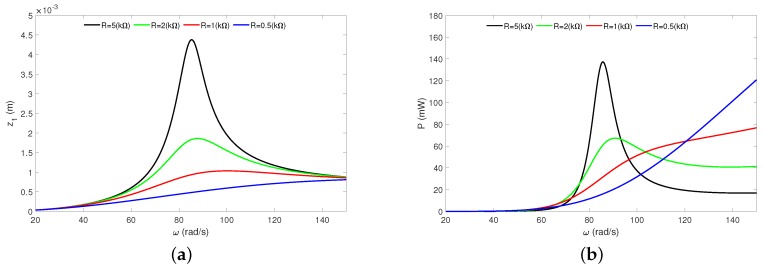
Analytical HBM results. Magnet’s resonance curves (**a**) and recovered power (**b**) under the resistance influence. The results have been calculated for amplitude *Q* = 0.6 mm and have obtained from Equations (Equation 27) and (Equation 28). All resonance curves are stable.

**Figure 9 sensors-20-01623-f009:**
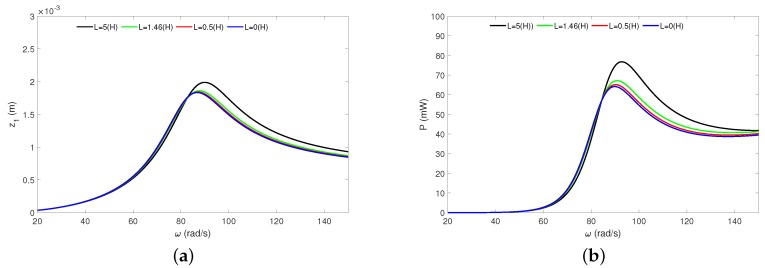
Analytical HBM results. Magnet’s resonance curves (**a**) and the recovered power (**b**) under the coil inductance influence. The results have been calculated for amplitude *Q* = 0.6 mm and the resistance *R* = 2 kΩ. The results have been obtained from Equations (Equation 27) and (Equation 28). All resonance curves are stable.

**Figure 10 sensors-20-01623-f010:**
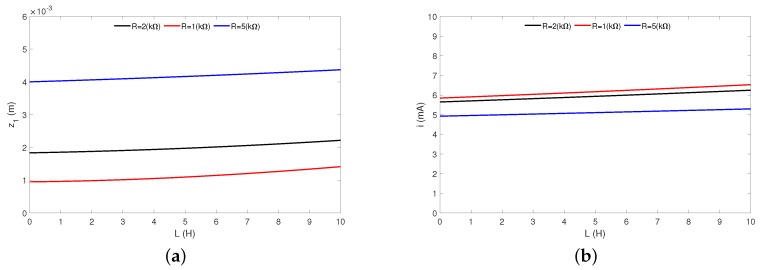
Influence of the coil inductance *L* on the magnet’s response (**a**) and the recovered current (**b**). The results have been obtained for frequency ω = 88 rad/s.

**Figure 11 sensors-20-01623-f011:**
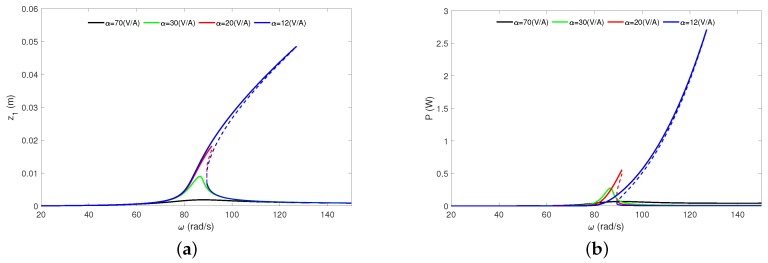
Analytical HBM results. Magnet’s resonance curves (**a**) and the recovered power (**b**) under the electromechanical coupling coefficient. The results have been calculated for *Q* = 0.6 mm and *R* = 2 kΩ. The results have been obtained from Equations (Equation 27) and (Equation 28). The dashed line means unstable solution obtained from HBM.

**Figure 12 sensors-20-01623-f012:**
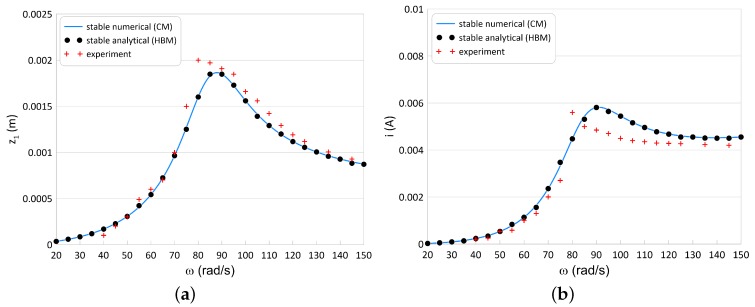
Magnet’s resonance curve (**a**) and the recovered current (**b**) for *R* = 2 kΩ and α = 70 V/A. The marked black points denote the HBM results, the blue lines mean numerical results and the plus signs are experimental results.

**Figure 13 sensors-20-01623-f013:**
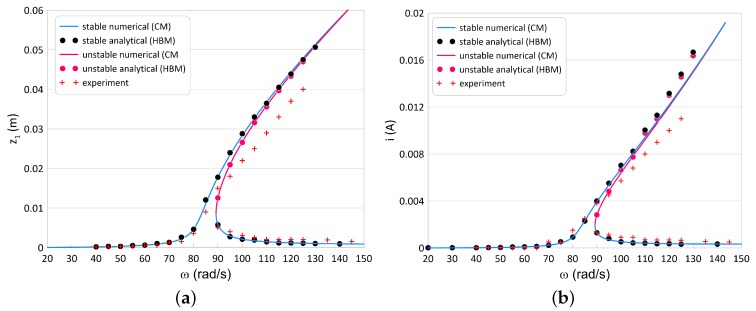
Magnet’s resonance curve (**a**) and the recovered current (**b**) for *R* = 10 kΩ and α = 25 Vs/m. The marked black points denote the HBM results, the blue and red lines mean numerical results and the plus signs are experimental results. The foldover effect toward high frequencies is observed. This is commonly named hardening.

**Figure 14 sensors-20-01623-f014:**
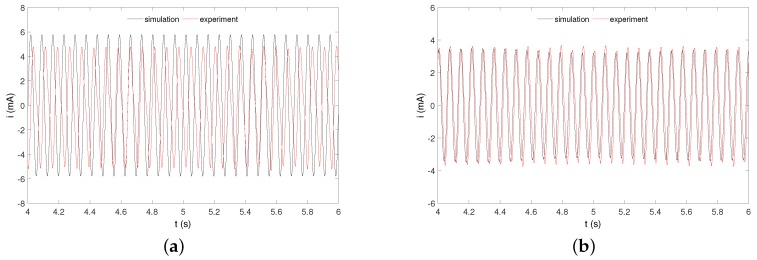
Experimental and numerical recovered currents obtained for parameters: α = 70 V/A, *R* = 2 kΩ (**a**) and α = 25 V/A, *R* = 10 kΩ (**b**). The frequency and amplitude of excitation were ω = 88 rad/s and *Q* = 0.6 m.

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
