# Peer review of "Theoretical and Experimental Investigations of a Pseudo-Magnetic Levitation System for Energy Harvesting"

_sensors, 2020, doi:10.3390/s20061623_

Round 1

Reviewer 1 Report

The paper presents analytical, numerical and experimental analysis of the special designed 1 system for energy harvesting. The harvester system consists of two identical magnets rigidly mounted 2 to the tube’s end. The comments are listed as follows

The energy harvesting efficiency should be calculated in the paper to show the energy harvesting performance. The paper should compare their havresting model to some tradtional harvesting model. The energy harvesting distance should be indicated in the paper. How to avoid the energy  leakage in proposed model. some relative work should be described such as:Liu Xin, Li Feng, Na Zhenyu. Optimal Resource Allocation in Simultaneous Cooperative Spectrum Sensing and Energy Harvesting for Multichannel Cognitive Radio. IEEE ACCESS, 2017, 5, 3801-3812ï¼›Liu Xin, Zhang Xueyan, Jia Min. 5G-based green broadband communication system design with simultaneous wireless information and power transfer. Physical Communication. 2018, 25, 539-545

Author Response

The authors would like to thank the editor for their consideration, and the reviewers for the time spent on carefully reviewing this work and for their valuable deep insight and comments. The authors feel that this paper is now clearer, more thoroughly discussed and better-referenced. The revised manuscript has been reviewed by a language expert. The work has been revised to address the reviewers’ suggestions. Please find hereafter a point-by-point reply to the comments and suggestions. Red words indicate changes from the original text of the manuscript.

The detailed answers are given in the attachment.

Reviewer 2 Report

The scope of this study is currently a significant research area. The research on electromagnetic energy harvesters using magnetic levitation architectures is a hot topic. This research area will most likely conduct to great impacts in the coming years.

The manuscript is well organized and introduces new findings. Nevertheless, I suggest some minor revisions before publication.

- A proofreading is required to remove many typos.

- Separate the scalar number from the si unit. Example:  w = 88 rad/s.

- Section 1, lines 19-21: I think you forgot to include the triboelectric harvesting. I recommend to cite the following papers:

Vladislav Slabov, et al. (2020). "Natural and Eco-Friendly Materials for Triboelectric Energy Harvesting", Nano-Micro Letters, 12:42.

João P. G. Tarelho, et al. (2018). "Graphene-based materials and structures for energy harvesting with fluids - A review", Materials Today, 21(10), 1019-1041.

- Section 1.1, Line 29: I think you must cite a paper focused on smart implant, as they require energy harvesting to supply sensors, actuators and wireless communication system. I propose the authors to cite the following papers:

Marco P. Soares dos Santos, et al. (2019). "Capacitive technologies for highly controlled and personalized electrical stimulation by implantable biomedical systems", Scientific Reports, 9: 5001.

Rodrigo Bernardo, et al. (2019). "Novel magnetic stimulation methodology for low-current implantable medical devices", Medical Engineering and Physics, 73, 77-84.

- Section 1.2: There is some relevant literature reviews in this scope that you can use to provide a stronger support of your sentences. I propose the authors to read and cite the most recent literature review paper, which I think it is the following one:

Pedro Carneiro, et al.(2020). "Electromagnetic energy harvesting using magnetic levitation architectures: a review", Applied Energy, 260: 114191.

- Section 1.2: it is not clear what is the problem that you propose to solve. It is not enough to the reader just a sentence stating: «Note that the detail analytical analysis have not yet been discussed in the literature». Please insert a very clear sentence stating the main problem being addressed in this paper and the main scientific advance achieved.

- Section 2.1: Use the categorization provided by Pedro Carneiro (2020) to detail the design configuration of your harvester.

- Section 2.2: Explain why the use of the model developed by Mann and Sims (2009).

- Section 3.1: Provide the assumptions you consider to use equations (3) and (4).

- Section 4.1: I think the content of this section must not be included in the section ‘Results”. It is suitable if included into a section “Methods”. As you do not have such a section “Methods”, choose a suitable section to insert it.

Author Response

(The authors gave the same response as above.)

Reviewer 3 Report

The submitted paper deals about an energy harvesting device to recover energy from environmental mechanical vibrations. The paper is clear and well written. The authors’s contributions are properly highlighted and the conclusions are widely supported by the analytical and experimental procedures.

However, I have some comments:

  • Line 35: please, replace “by modification…” with “by modifying…” or “by modifications of the…”;
  • Line 41: “According to theorem due to Earnshaw proves…” should be rewritten, eg as “The Earnshaw’s theorem proves…”
  • Line 51: please, substitute “modelling” with “modelled”.
  • Lines 51-52: Please, reformulate correctly the sentence.
  • Figure 2 is not properly explained. Please, insert more information or delete it.
  • Line 64: please, replace initial “the paper…” with “this paper…”;
  • Line 80: “inner tube’s surface”
  • Line 83: please replace “the high induction” with “the inductance of”;
  • Please, explain that g in eq. 1 is the gravitational acceleration.
  • Line 110 and figure 3: It is not clear how the acceleration sensor is located and how it can measure the magnet’s response.
  • 27 and 28 please, explicitate the term z1.
  • Line 146: “Therefore, the magnet vibration centre must take into consideration.”…not clear, perhaps is it “Therefore, the magnet vibration centre must be taken into consideration.” ?
  • Line 169: it is written “the load resistance R”, but previously R has been defined as the total resistance. This is misleading. Also, in the following text and figures it is not clear…it seems that the total resistance has been varied, instead of the load resistance. I suggest to modify the following sentence and figure, in order to use only RL which is the real circuital parameter (indeed RC is fixed once the coil is chosen).
  • Figure 14 and 15, please reduce the range time, in order to better appreciate the waveforms comparison.
  • Finally, regarding the sentence in line 235 about the noise, I would like to suggest to increase the distance among the harvester device and the shaker moving surface, in order to reduce the electromagnetic coupling of the sensing coil with the power coil in the shaker (this is often done in cantilever beam harvester based on magnetostrictive or piezo materials). For example, by increasing the height of the brass bar.

Author Response

(The authors gave the same response as above.)

Round 2

Reviewer 1 Report

The paper can be accepted now